# Diet Quality, Microbial Lignan Metabolites, and Cardiometabolic Health among US Adults

**DOI:** 10.3390/nu15061412

**Published:** 2023-03-15

**Authors:** Nicholas A. Koemel, Alistair M. Senior, Tarik Benmarhnia, Andrew Holmes, Mirei Okada, Youssef Oulhote, Helen M. Parker, Sanam Shah, Stephen J. Simpson, David Raubenheimer, Timothy P. Gill, Nasser Laouali, Michael R. Skilton

**Affiliations:** 1Charles Perkins Centre, The University of Sydney, Sydney 2006, Australia; 2Sydney Medical School, The University of Sydney, Sydney 2006, Australia; 3School of Life and Environmental Sciences, The University of Sydney, Sydney 2006, Australia; 4Sydney Centre for Precision Data Science, The University of Sydney, Sydney 2006, Australia; 5Scripps Institution of Oceanography, University of California, San Diego, CA 92093, USA; 6Department of Biostatistics and Epidemiology, School of Public Health and Health Sciences, University of Massachusetts, Amherst, MA 01003, USA; 7“Exposome and Heredity” Team, CESP, Paris-Saclay University, UVSQ, University Paris-Sud, Inserm, Gustave Roussy, F-94805 Villejuif, France; 8Susan Wakil School of Nursing and Midwifery, The University of Sydney, Sydney 2006, Australia

**Keywords:** diet quality, Healthy Eating Index, microbiome, gut health, cardiometabolic health, enterolactone, enterodiol, NHANES

## Abstract

The gut microbiome has been shown to play a role in the relationship between diet and cardiometabolic health. We sought to examine the degree to which key microbial lignan metabolites are involved in the relationship between diet quality and cardiometabolic health using a multidimensional framework. This analysis was undertaken using cross-sectional data from 4685 US adults (age 43.6 ± 16.5 years; 50.4% female) participating in the National Health and Nutrition Examination Survey for 1999–2010. Dietary data were collected from one to two separate 24-hour dietary recalls and diet quality was characterized using the 2015 Healthy Eating Index. Cardiometabolic health markers included blood lipid profile, glycemic control, adiposity, and blood pressure. Microbial lignan metabolites considered were urinary concentrations of enterolignans, including enterolactone and enterodiol, with higher levels indicating a healthier gut microbial environment. Models were visually examined using a multidimensional approach and statistically analyzed using three-dimensional generalized additive models. There was a significant interactive association between diet quality and microbial lignan metabolites for triglycerides, low-density lipoprotein cholesterol, high-density lipoprotein cholesterol, insulin, oral glucose tolerance, adiposity, systolic blood pressure, and diastolic blood pressure (all *p* < 0.05). Each of these cardiometabolic health markers displayed an association such that optimal cardiometabolic health was only observed in individuals with both high diet quality and elevated urinary enterolignans. When comparing effect sizes on the multidimensional response surfaces and model selection criteria, the strongest support for a potential moderating relationship of the gut microbiome was observed for fasting triglycerides and oral glucose tolerance. In this study, we revealed interactive associations of diet quality and microbial lignan metabolites with cardiometabolic health markers. These findings suggest that the overall association of diet quality on cardiometabolic health may be affected by the gut microbiome.

## 1. Introduction

Cardiometabolic risk factors such as hypertension, elevated fasting blood sugar, dyslipidemia, and abdominal obesity have increased in prevalence over the past two decades [1,2]. Collectively, cardiometabolic disorders were responsible for more than 4.8 million deaths among the US working-age (ages 25–64) population between 1990 and 2017 [3]. A spectrum of modifiable risk factors, including environment, lifestyle, and diet, have been identified for cardiometabolic disorders. From a nutrition standpoint, there are many components of diet quality that can impact cardiometabolic health, including fiber, sodium, fatty acids, added sugars, polyphenols, and antioxidants [4]. Adherence to higher overall diet quality has been shown to improve overall cardiometabolic health [5,6].

It is well established that diet quality and microbial metabolism interact to influence multiple processes relevant to cardiometabolic health [7,8]. The gut microbiota is involved in the production and release of metabolites to systemic tissue, extraction of nutrients, synthesis of specific vitamins, alteration of gastrointestinal hormones, and nerve function [9,10,11]. Microbial metabolites have been further implicated in host metabolic regulation of inflammation [12], lipid metabolism [13], and type 2 diabetes risk [14]. Previous studies reported that the gut microbiome plays an important role in the protective effects observed from consuming healthy dietary patterns, such as a Mediterranean diet [13] or an anti-inflammatory diet [15]. Plant foods are rich sources of polyaromatic compounds via lignans and flavonoids found in their cell walls [16]. Lignans have been of particular interest as substances responsible for the beneficial effect of consuming nuts, fruits, vegetables, whole grains and overall plant-based diets. The gut microbiota plays an important role in this benefit by converting the dietary plant lignans to produce more bioactive enterolignans, such as enterolactone and enterodiol.

On the other hand, unhealthy foods containing high amounts of saturated fat, refined sugars, emulsifiers, and sodium have been shown to elicit negative effects on microbial health [17]. Diets containing large amounts of processed foods have been linked to lower microbial diversity [18], reduced abundance of beneficial taxa [19], and ultimately a lower capacity to produce cardioprotective microbial metabolites such as enterolignans [20]. The bidirectional association between both healthy (plant-based) and unhealthy dietary components with the gut microbiome suggests a complex interplay with cardiometabolic health. However, the degree to which overall diet quality, gut microbiota function, and cardiometabolic disorders interact has not been fully defined.

The National Health and Nutrition Examination Survey (NHANES) offers the opportunity to explore these relationships on a larger scale using a cross-sectional design. In the absence of microbial taxonomic composition or various other microbial metabolites, we explored enterolactone and enterodiol, which serve as a surrogate marker of gut microbiota function. Diet quality was evaluated using the Health Eating Index 2015 (HEI) in order to capture both healthy and unhealthy components of the diet. To better quantify and visualize the associations with cardiometabolic health, we applied a multidimensional approach that has previously demonstrated the ability to capture relationships not achievable by traditional univariate analyses [21,22,23]. In the present study, we hypothesized that microbial lignan metabolites would support a potential effect-modifying role of the gut microbiome on the relationship between diet quality and cardiometabolic health.

## 2. Materials and Methods

### 2.1. Study Population

This study examined data from the NHANES dataset collected annually in the US by the National Center for Health Statistics. This ongoing cross-sectional survey aims to assess the nutritional intake and overall health of those living in the US. NHANES data were collected from 1999 to 2010 for participants aged 20 and older. Dietary data were collected by a trained nutritional professional via two separate dietary recalls. An initial 24 h recall was collected during the in-person interview and the second recall was conducted 3–10 days later by telephone. Additionally, participants who reported having cardiovascular disease (n = 668), cancer (n = 511), diabetes (n = 523), or related medication (n = 4788), were excluded from the primary analysis (Appendix A).

### 2.2. Urinary Enterolignans

Enterolactone and enterodiol were measured from urine samples collected at the initial interview in those who had fasted a minimum of 9 h and immediately stored at −20 °C until processing. High-performance liquid chromatography was then used to quantify the concentration of enterolactone and enterodiol in the urine. Antibiotic consumption has the potential to influence urinary enterolignan concentration by destroying the intestinal microflora [24], so individuals who reported taking antibiotics within a month of the collected enterodiol or enterolactone sample were excluded (n = 9) [25]. Enterodiol and enterolactone values were log-transformed to address skewness.

### 2.3. The Healthy Eating Index

The most recent HEI was developed in 2015 to measure overall diet quality and presents a composite measure of conformance to the 2015–2020 Dietary Guidelines for Americans [26]. The HEI is a 100-point scale, with a higher score indicating better overall diet quality. The adequacy components include total fruit (5), whole fruits (5), total vegetables (5), greens and beans (5), whole grains (10), dairy (10), total protein foods (5), seafood and plant proteins (5), and fatty acids (ratio of the sum of polyunsaturated and monounsaturated fatty acids to saturated fatty acids—10). The moderation components include refined grains (10), sodium (10), added sugars (10), and saturated fats (10).

### 2.4. Cardiometabolic Health

A sample of participants was selected for measurement of fasting serum glucose, insulin, hemoglobin A1c (HbA1c), total cholesterol (Total-C), low-density lipoprotein (LDL) cholesterol, high-density lipoprotein (HDL) cholesterol, and triglycerides. An oral glucose tolerance test (OGTT) was administered using a calibrated dose of glucose drink (Trutol^TM^, Thermo Scientific, Waltham, MA, USA) providing on average 75 g of glucose. Postprandial glucose was measured 2 h after the consumption of the glucose drink. Body fat percentage was estimated via bioelectrical impedance. Systolic and diastolic blood pressure was measured 3–4 times via sphygmomanometer and the average of the measurements used in the analysis. Height and weight were collected by a trained professional following standardized operating procedures, with body mass index (BMI) calculated as weight divided by height in meters squared (kg/m^2^).

### 2.5. Demographic and Lifestyle Covariates

All demographic and lifestyle covariates were self-reported via questionnaires. Race and ethnicity were categorized as either non-Hispanic white, non-Hispanic black, Hispanic, or other. Level of education was categorized as less than high school, high school, or some college and above. Socioeconomic status was calculated using household income to poverty ratio. Participants were classified as smokers if they reported smoking >100 cigarettes in their lifetime. Alcohol consumption was categorized as “drinkers” and “nondrinkers,” where those who drank a minimum of 12 drinks within any given year prior to the assessment were considered drinkers. Physical activity was determined using self-reported metabolic equivalents of weekly moderate to vigorous leisure activity. In the case of a missing value, the mean value of the covariate was utilized.

### 2.6. Statistical Analysis

For the present study, individuals with potential over- or underreporting for dietary energy intake were excluded (males < 800 or >4200 kcal/day and females < 600 or >3500 kcal/day; n = 8087). Participants were also excluded if consuming macronutrients greater or fewer than three standard deviations from the mean (n = 387).

Associations of dietary quality, enterolactone, enterodiol, and cardiometabolic health markers were explored using generalized additive models (GAMs). GAMs are a dynamic form of multivariable regression that can be used to test for and visualize complex nonlinear associations [27]. Models include smooth terms that handle more complex dimensions of data and varying scales. For each cardiometabolic marker, a series of GAMs including additive and interactive associations was implemented to sequentially explore the complex relationship between HEI score, total energy intake, and either enterolactone or enterodiol (Appendix A). The most complex model contained a three-dimensional smooth term that included HEI score, total energy intake, and either enterolactone or enterodiol. Total energy intake was included in the smooth term as an adjustment approach [28] while simultaneously allowing for us to explore visual differences at varying energy intakes. A series of models was then designed to sequentially adjust for confounding variables as additive terms. Model one was adjusted for age, sex, and socioeconomic status. Model two further was adjusted for sociodemographic characteristics, including race/ethnicity and education. Model three was the fully adjusted model and further adjusted for lifestyle factors, such as alcohol consumption, smoking, BMI, and physical activity. All models were constructed using the “gam” function of the *mgcv* package in R statistical software (v. 1.8–41; R Core Team; Vienna, Austria) [29,30].

Associations were visualized using three-dimensional response surfaces, where each cardiometabolic health marker was plotted as a response surface at the 25th, 50th, and 75th percentile of total energy intake. Response surfaces show the outcome on a scale with warmer colors denoting higher values and cooler colors denoting lower values. A statistically significant three-dimensional term for the exposures of interest can be interpreted such that the association between HEI with cardiometabolic health depends on the urinary enterolignan and total energy intake. To explore the degree to which cardiometabolic health outcomes are related to HEI across a spectrum of microbial lignan metabolite levels, we display the response surfaces at the 50th percentile of total energy intake. Additional figures are provided in the online supplement presenting the associations at the 25th, and 75th percentile of total energy intake. Values were estimated using generalized crossed validation and checked for overfitting. Interaction between sex with each biomarker was explored using the “by” term in the “gam” function of *mgcv*. The Akaike information criterion (AIC) was used as a measure for model comparison, with lower values indicating better fit relative to the increase in model complexity. A difference in AIC > 2 was considered evidence of a better overall model fit [31]. Sex-stratified analyses were also undertaken for each of the cardiometabolic health markers. A sensitivity analysis using waist circumference instead of BMI was conducted to act as a better indicator of central adiposity.

## 3. Results

### 3.1. Participant Characteristics

Participant characteristics are presented in Table 1. This analysis included 4685 US adults (43.6 ± 16.5 years; 50.4% female). Participants were predominantly non-Hispanic white (46.6%) and overweight with an average BMI of 28.5 ± 6.5 kg/m^2^.

### 3.2. Generalized Additive Model Exploration

The series of potential interactive and additive models we used for exploring cardiometabolic health markers is shown in Appendix A. Marginal differences were observed when comparing deviance explained and AIC values for the various models. Notably, triglycerides and oral glucose tolerance test were the only biomarkers that favored the more complex three-way interactive model.

### 3.3. Blood Lipids

The association of microbial lignan metabolites and HEI with blood lipids at the 50th percentile of energy intakes is shown in Figure 1 and model coefficients are displayed in Table 2. Results at the 25th and 75th percentile of energy intakes are displayed in Appendix A. There was a statistically significant association with enterolactone and enterodiol displayed for triglycerides (all *p* ≤ 0.002), LDL cholesterol (all *p* ≤ 0.04), and HDL cholesterol (all *p* < 0.001). Urinary enterolactone levels appeared to be inversely associated with plasma triglycerides in people with HEI above the mean, with the highest triglyceride levels being present at <1 μmol/L (log-transformed) enterolactone and the lowest triglycerides at around 4 μmol/L (log-transformed). Enterodiol showed a potential interactive association with HEI, evidenced by the bending contour lines particularly prominent at higher levels of HEI and enterodiol (Figure 1A).

These associations appeared similar across energy intakes, albeit that higher overall triglycerides were observed at the 25th percentile of energy intake (Appendix A). LDL cholesterol was primarily associated with HEI for both enterolactone and enterodiol (Figure 1B). Across energy intakes, the association remained similar, but the lowest LDL cholesterol values were observed with higher energy intake coupled with higher HEI and enterodiol. The inverse was apparent for enterolactone, where the lowest LDL cholesterol was evident in those with the lowest energy intake coupled with the highest HEI and enterolactone (Appendix A). Visually, HDL cholesterol followed an interactive association for both enterolactone and enterodiol (Figure 1C). This appeared similar across energy intakes, although at the upper level of energy intake, HEI appeared to have a slightly stronger positive association with HDL (Appendix A). We did not identify any association with total cholesterol in the fully adjusted model (Appendix A and Table 2). When adjusting for waist circumference instead of BMI, all associations remained the same except for LDL and enterolactone, which become nonsignificant (Appendix A).

Male and female stratified results for cardiometabolic health markers are shown in Appendix A; Appendix A. When formally comparing the AIC values, there was strong support of a sex difference between blood lipid models. We found a positive association for enterodiol and triglycerides for both males and females (*p* ≤ 0.001). Visually, males and females had a similar response surface to the pooled analysis for the association of enterolactone with triglycerides; however, this only reached statistical significance in females (*p* = 0.008). HDL cholesterol was statistically significant in both males and females (*p* < 0.001), with little difference in the response surface compared to the pooled analysis. Evidence for an interaction with sex was revealed for triglycerides (Appendix A).

### 3.4. Glycemic Control

The relationship between energy intake, microbial lignan metabolites, and HEI with markers of glycemic control is shown in Figure 2 and model coefficients are displayed in Table 2. Enterolactone revealed a weak interactive association with HEI for fasting insulin levels (Figure 2A). The highest fasting insulin appeared in those with the lowest HEI and enterolactone levels. Enterodiol associations were more complex with the highest insulin at low HEI and enterodiol, but this became less apparent beyond an enterodiol of 2 μmol/L (log-transformed). These associations also displayed differences across energy intake, where the highest fasting insulin was observed at lower energy intake coupled with lower HEI and lower enterodiol. At higher energy intake, the association became more HEI dominated. For enterolactone, the highest fasting insulin was apparent at higher energy intakes, but at lower HEI and lower overall enterolactone (Appendix A). In the fully adjusted model, OGTT was significantly associated with enterodiol (*p* = 0.03) and had a near-significant association with enterolactone (*p* = 0.08). Enterolactone and enterodiol both displayed a strong negative association with OGTT responses, with near-vertical contour lines across the enterodiol and enterolactone spectrum (Figure 2B). For enterodiol and enterolactone, the overall association was stronger at lower energy intakes (Appendix A). There was no significant association identified for fasting glucose or HbA1c (Appendix A; Table 2). For waist circumference sensitivity, all associations remained the same except the association between OGTT and enterodiol, which became nonsignificant (Appendix A).

In the sex-stratified analysis, there was strong support for a sex difference between models of glycemic control comparing the model AIC values (Appendix A; Appendix A). Fasting glucose was significant only with enterodiol in males (*p* = 0.03). The response surface revealed a robust negative association between enterodiol levels and fasting glucose independently of HEI. Males had a significant association with HbA1c, but only for enterodiol (*p* = 0.007). Females had a significant association with both enterolactone and enterodiol with insulin (*p* < 0.001), with response surfaces suggesting a strong negative association with HEI. There was no evidence of an interaction by sex in any of the markers of glycemic control (Appendix A).

### 3.5. Adiposity and Blood Pressure

The relationship between energy intake, microbial lignan metabolites, HEI with adiposity and blood pressure is shown in Figure 3 and model coefficients are displayed in Table 2. Adiposity had a significant interactive association with HEI and both enterodiol (*p* = 0.02) and enterolactone (*p* = 0.007). Adiposity was lowest in participants who had high HEI in combination with high levels of enterolactone (Figure 3A). For enterodiol, adiposity had a stronger HEI association, where the lowest adiposity was observed in those with the highest HEI (Figure 3B). At lower energy intakes, higher adiposity was observed in individuals with low enterolactone despite higher HEI scores. However, at higher energy intake, the association appears to become slightly more HEI-dominated (Appendix A). A significant interactive association was detected for both enterodiol and enterolactone with systolic blood pressure (*p* < 0.001). Visually, systolic blood pressure was highest in those with low HEI and low enterolactone or low enterodiol. Diastolic blood pressure was also significantly associated with enterolactone (*p* = 0.005) and enterodiol (*p* = 0.007), displaying a near-identical relationship, as seen with systolic blood pressure (Figure 3C). Both systolic and diastolic blood pressure showed similar associations across energy intakes, but with a stronger association with enterodiol and enterolactone at lower energy intakes (Appendix A). There were no significant changes for adiposity and blood pressure in the waist circumference sensitivity analysis (Appendix A).

There was strong support of a sex difference between models of adiposity and blood pressure when comparing AIC values (Appendix A; Appendix A). Systolic blood pressure was significantly associated with both enterolactone (*p* = 0.003) and enterodiol for males (*p* = 0.04). In females, only enterodiol was significantly associated with systolic blood pressure (*p* = 0.04). Only females demonstrated a significant association with diastolic blood pressure and microbial lignan metabolites (enterolactone: *p* = 0.04; enterodiol: *p* = 0.03). Both markers visually displayed an interactive association similar to the pooled analyses. There was no evidence of an interaction by sex for adiposity or blood pressure (Appendix A).

## 4. Discussion

Using a large sample of US adults, we explored the potential associations between diet quality and microbiome lignan metabolites with cardiometabolic health across low, medium, and high levels of energy intake using three-dimensional visualization. Across all energy intake levels, gut microbiome metabolites, and the HEI were interactively associated with most cardiometabolic markers evaluated in this study. Generally, we found that higher levels of enterodiol or enterolactone in combination with greater adherence to the HEI were associated with more optimal cardiometabolic health.

Numerous studies have provided evidence that diet quality plays a role in cardiometabolic health [32,33]. The HEI provides a measure of adherence to the Dietary Guidelines for Americans and encompasses multiple dimensions of diet quality, including high-quality plant-based food items and dietary components related to unhealthy foods. The HEI emphasizes a higher consumption of fruit, vegetables, whole grains, and nuts and legumes while limiting sodium, refined grains, added sugar, and saturated fat [26]. Adherence to a diet with a high HEI score is associated with protective effects against obesity, diabetes mellitus, dyslipidemia, and hypertension [34,35,36].

Experimental evidence supports that the relationship between both beneficial [13] and detrimental components [32] of diet quality with cardiometabolic health is partially mediated through the gut microbiome. Unlike the complex interplay identified in this analysis, various studies have investigated the independent role of the diet or the gut microbiome on cardiometabolic health [37]. An altered gut microbiome composition has been well documented to influence the development of metabolic disorders such as obesity, diabetes mellitus, dyslipidemia, and hypertension [38,39,40]. The potential mechanisms have been summarized recently by Kazemian et al. [41]. Such an association can be through indirect (via the immune system) and direct (via metabolites such as enterodiol and enterolactone) pathways [39,40]. Microbial lignan metabolites have several biological functions, such as antioxidant and ligand activity [42]. This includes increasing hepatic LDL cholesterol receptor activity [43] and acting as an antagonist of platelet-activating factor [44]. Together, these metabolites provide several potential mechanisms for reducing the risk of cardiometabolic diseases.

Understanding the interplay between diet and gut microbiome metabolites on cardiometabolic diseases is of public health and clinical importance. These results highlight this importance by demonstrating the potential magnitude to which the gut microbiome may modify the relationship between diet quality and cardiometabolic health outcomes. Moreover, gut health may be an influential characteristic to consider when aiming to optimize cardiometabolic health with dietary modifications. In line with our results, Asnicar et al. revealed numerous relationships between microbes, dietary nutrients, and several dietary indices, suggesting that the microbiome modulates the effect of the diet on both fasting and postprandial cardiometabolic health [45]. Moreover, a recent study on overall dietary lignan intake and cardiometabolic risk in men (n = 911) reported that both gut microbial species and plasma enterolactone levels accounted for a significant proportion of the association observed [46]. Of the relationship between dietary lignan intake and metabolic health, microbial species alone explained 19.8% (95% CI: 7.3–43.6%), while species and enterolactone levels collectively explained 54.5% (95% CI: 21.8–83.7%) of the relationship. The interplay of diet quality and gut microbiome metabolites on cardiometabolic health is biologically plausible, as diet may potentially modulate production of gut microbiome metabolites by altering the gastrointestinal microbiota composition. Previous studies have shown an increased gut microbial diversity among people with higher fiber intake [47]. In contrast, digestible simple sugars inhibit the colonization of beneficial commensal microbial species in the murine gut and promote the development of obesity [48].

Our findings also identified the potential of sex-specific differences. Specifically, response surfaces were slightly different for females and males, suggesting the beneficial associations between enterolactone and enterodiol with cardiometabolic markers may be more pronounced in males compared to females. These findings accord with previous studies that demonstrate an influential effect of biological sex on the physiology and pathology of cardiometabolic diseases [49]. Other studies have also reported sex differences in the association between gut microbiome and cardiometabolic disease [50]. The mechanisms are not fully understood, although studies suggest a complex bidirectional interaction between the microbial community and sex hormones [51].

### Strengths and Limitations

This study has several strengths, including the analysis of a large sample of US adults and the use of objective laboratory values of urine enterolignan levels and serum cardiometabolic marker measurements. To our knowledge, this is the first study to incorporate a multidimensional framework to visualize the relationship between diet, microbial lignan metabolites, and cardiometabolic health. Unlike traditional epidemiological approaches, this technique enables us to visually capture the complex relationships and how they differ in magnitude for each cardiometabolic marker. However, this study also has several limitations. The complex NHANES survey design weights could not be applied in this study because the R package used does not allow it, thus preventing these results from being generalized to the entire US population. However, it has been suggested that weighted analyses can be inefficient due to the large variability in assigned weights [52]. The unweighted analysis can yield correct estimates when models are adjusted for the auxiliary variables used to define the weights (i.e., age, sex, and ethnicity). Another limitation is the cross-sectional design of NHANES such that the results cannot support causal inferences about the relationships between diet, gut microbiome metabolites, and cardiometabolic health. In addition, reverse causality is possible given the cross-sectional design. As discussed, several previous longitudinal studies have demonstrated individual associations between adherence to the HEI with enterodiol and enterolactone and cardiometabolic diseases. Diets with a higher HEI score often include more plant-based food items with a greater overall lignin content. Furthermore, the relationship between HEI and lignin-containing food items may have partially influenced some of the observed results. In addition, the dietary consumption data used to calculate the HEI was collected via 24 h recalls and may not represent the usual dietary intake of individuals, as under- or overreporting frequently occurs. Lastly, this study did not consider the consumption of dietary supplements.

## 5. Conclusions

This study applied a novel multidimensional approach to explore the relationship between diet quality, microbial lignan metabolites, and cardiometabolic health among US adults. We revealed that enterolactone and enterodiol affect the relationship of diet quality with blood lipids, glycemic control, adiposity, and blood pressure. Future research is needed to explore what specific foods or dietary patterns may underpin this relationship.

## Figures and Tables

**Figure 1 nutrients-15-01412-f001:**
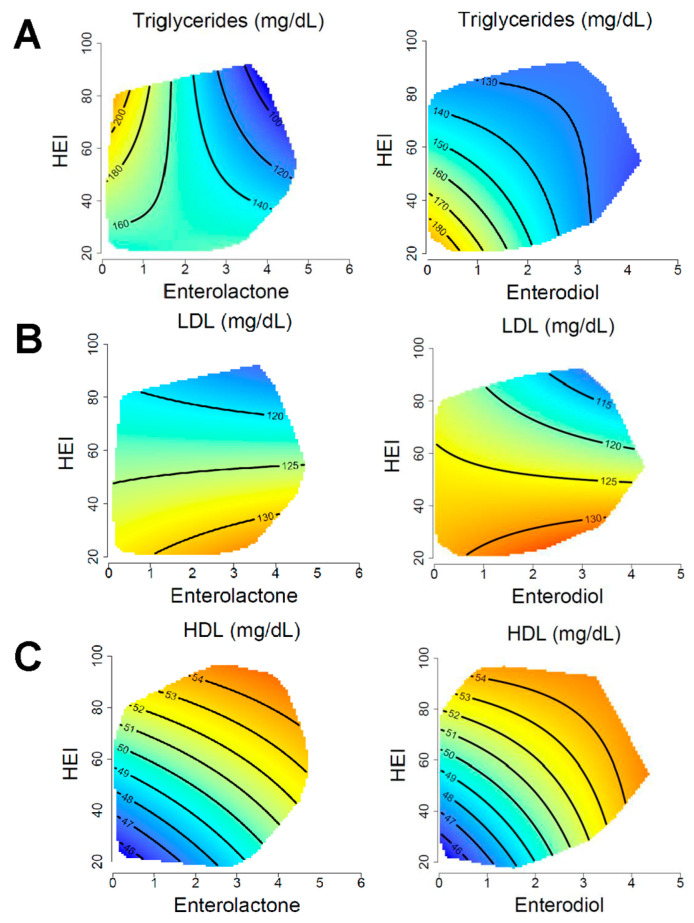
Associations of blood lipids with HEI and enterolignans. Response surfaces show the association of triglycerides (**A**) LDL cholesterol (**B**) and HDL cholesterol (**C**) with HEI, enterolignans, and total energy intake. Enterolactone and enterodiol are presented as μmol/L (log-transformed). The outcome of each response surface is shown at the top of the plot, with warmer colors denoting higher values and cooler colors denoting lower values. Response surfaces are predicted at the 50th percentile of total energy intake and have been adjusted for age, sex, household income, BMI, physical activity. Low-density lipoprotein (LDL); high-density lipoprotein (HDL).

**Figure 2 nutrients-15-01412-f002:**
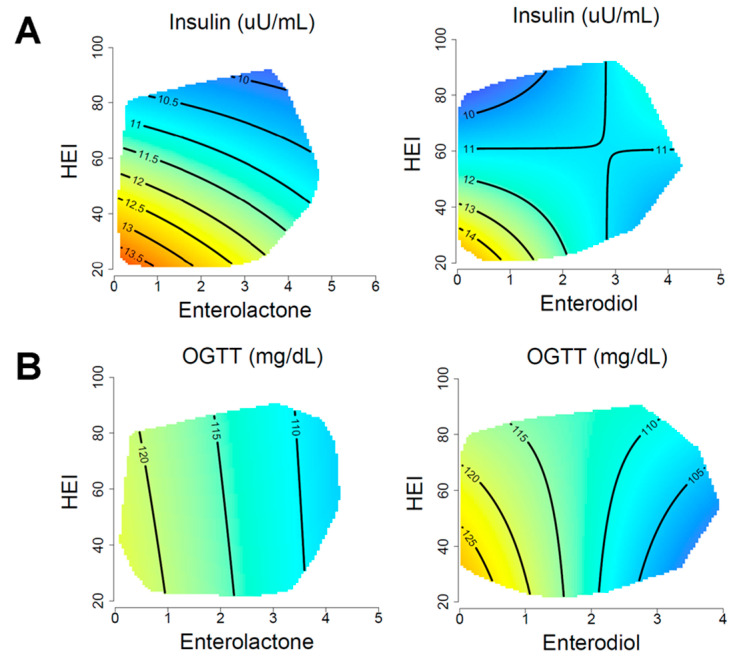
Associations of glycemic control with HEI and enterolignans. Response surfaces show the association of insulin (**A**) and OGGT (**B**) with HEI, enterolignans and total energy intake. Enterolactone and enterodiol are presented as μmol/L (log-transformed). The outcome of each response surface is shown at the top of the plot, with warmer colors denoting higher values and cooler colors denoting lower values. Response surfaces are predicted at the 50th percentile of total energy intake and have been adjusted for age, sex, household income, BMI, physical activity. Oral glucose tolerance test (OGTT).

**Figure 3 nutrients-15-01412-f003:**
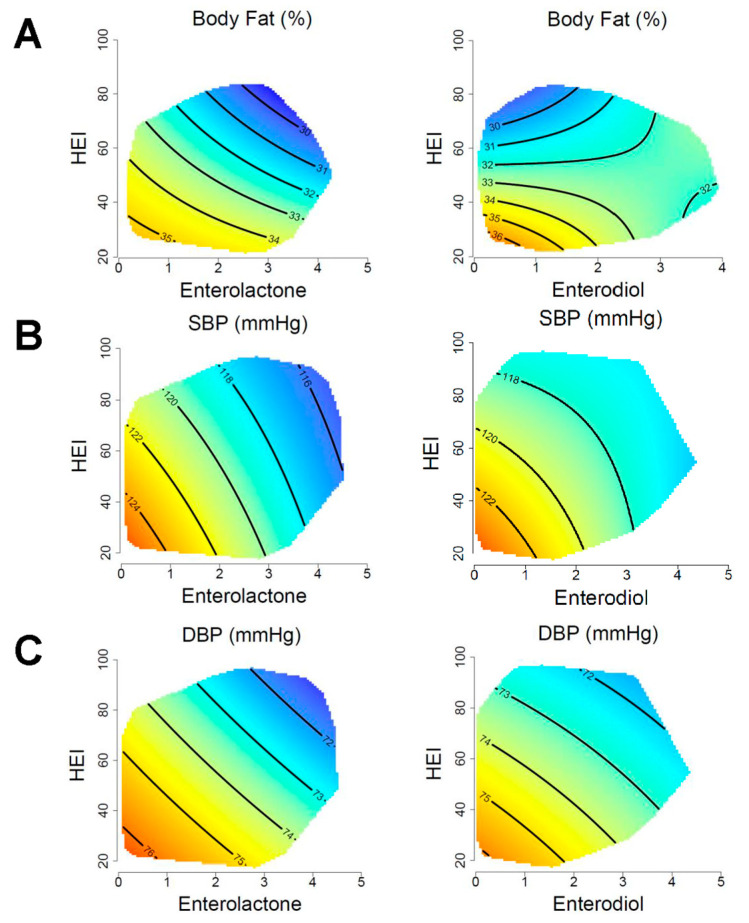
Associations of adiposity and blood pressure with HEI and enterolignans. Response surfaces show the associations of adiposity (**A**), systolic blood pressure (**B**), and diastolic blood pressure (**C**) with HEI, enterolignans, and total energy intake. Enterolactone and enterodiol are presented as μmol/L (log-transformed). The outcome of each response surface is shown at the top of the plot, with warmer colors denoting higher values and cooler colors denoting lower values. Response surfaces are predicted at the 50th percentile of total energy intake and have been adjusted for age, sex, household income, BMI, and physical activity.

**Table 1 nutrients-15-01412-t001:** Participant Characteristics ^1^.

Participant Characteristics	Mean	SD
Age (years)	43.6	16.5
Female sex (%)	50.4	−
BMI (kg/m^2^)	28.5	6.5
Total Energy (kcal)	2101	737
Healthy Eating Index Score	50.9	12.1
Enterolactone μmol/L (log-transformed)	2.54	0.75
Enterodiol μmol/L (log-transformed)	1.59	0.64
Protein (TEI%)	15.0	3.4
Carbohydrate (TEI%)	51.5	7.7
Fat (TEI%)	33.4	6.6
Fiber (g)	15.6	7.8
Sugar (g)	78.0	7.5
Sodium (mg)	2125	1256.4
Race/Ethnicity		
Hispanic (%)	30.3	−
Non-Hispanic White (%)	46.6	−
Non-Hispanic Black (%)	19.0	−
Other (%)	4.1	−
Family Income to Poverty Ratio	2.54	
Education Level		
Less than high school (%)	28.1	−
High school graduate or GED (%)	23.2	−
Some College or More (%)	48.7	−
Nondrinker (%)	24.6	−
Nonsmoker (%)	49.5	−
Physical Activity (METs)	1811	2363
Lipid Profile		
Triglycerides (mg/dL)	135.3	113.3
Total Cholesterol (mg/dL)	200.3	40.7
LDL Cholesterol (mg/dL)	120.2	34.3
HDL Cholesterol (mg/dL)	52.2	15.7
Glycemic Control		
Glucose (mg/dL)	99.2	21.6
Insulin (uU/mL)	12.1	9.7
OGTT (mg/dL)	114.8	50.9
HbA1c (%)	5.43	0.67
Adiposity and Blood Pressure		
Body Fat (%)	31.9	10.71
Systolic Blood Pressure (mmHg)	121.3	17.6
Diastolic Blood Pressure (mmHg)	70.9	11.4

^1^ Participant Characteristics. Body mass index (BMI); percentage of total energy intake (TEI%); standard deviation (SD). Low-density lipoprotein (LDL); high-density lipoprotein (HDL); oral glucose tolerance test (OGTT); hemoglobin A1c (HbA1c).

**Table 2 nutrients-15-01412-t002:** Model coefficients for cardiometabolic health, diet quality, and microbial lignan metabolites ^1^.

		Model 1			Model 2			Model 3		
Outcome	Metabolite	DE	AIC	p	DE	AIC	p	DE	AIC	p
Triglycerides	Enterodiol	10.1%	25,710.0	<0.001	12.5%	25,666.9	0.002	18.2%	25,536.0	0.002
	Enterolactone	10.8%	25,684.1	<0.001	13.6%	25,624.2	<0.001	19.1%	25,489.3	<0.001
Total Cholesterol	Enterodiol	11.9%	40,395.2	0.31	12.0%	40,397.7	0.29	14.0%	40,319.9	0.43
	Enterolactone	12.0%	40,394.7	0.27	14.0%	40,319.9	0.43	14.0%	40,318.2	0.31
LDL Cholesterol	Enterodiol	10.5%	21,114.5	0.007	10.5%	21,118.0	0.006	13.3%	21,068.4	0.01
	Enterolactone	10.4%	21,116.9	0.02	10.4%	21,120.2	0.01	13.2%	21,071.1	0.04
HDL Cholesterol	Enterodiol	13.8%	36,135.9	<0.001	15.3%	36,048.6	<0.001	26.0%	35,463.1	<0.001
	Enterolactone	13.9%	36,130.5	<0.001	15.3%	36,057.7	<0.001	25.9%	35,466.2	<0.001
Glucose	Enterodiol	12.7%	19,190.4	0.046	13.6%	19,166.9	0.12	17.0%	19,089.1	0.24
	Enterolactone	12.4%	19,201.6	0.13	13.4%	19,175.7	0.25	16.8%	19,097.3	0.47
Insulin	Enterodiol	2.8%	15,052.2	<0.001	3.6%	15,049.0	<0.001	36.1%	14,016.4	0.02
	Enterolactone	3.8%	15,030.0	<0.001	4.6%	15,029.6	<0.001	36.3%	14,014.4	0.002
OGTT	Enterodiol	16.3%	10,335.4	0.03	17.4%	10,324.7	0.04	23.2%	10,263.5	0.03
	Enterolactone	16.2%	10,342.1	0.046	17.2%	10,331.1	0.07	22.9%	10,272.9	0.08
HbA1c	Enterodiol	10.3%	4879.4	0.38	11.2%	4851.0	0.51	15.9%	4700.1	0.28
	Enterolactone	10.5%	4869.2	0.11	11.4%	4841.9	0.19	15.6%	4713.7	0.92
Body Fat (%)	Enterodiol	43.9%	7948.2	0.03	43.8%	7950.7	0.02	44.6%	7940.3	0.02
	Enterolactone	44.0%	7946.0	0.01	43.9%	7948.3	0.007	44.7%	7938.1	0.007
Systolic Blood Pressure	Enterodiol	29.4%	36,898.7	<0.001	29.6%	36,889.1	<0.001	31.8%	36,767.0	<0.001
	Enterolactone	29.9%	36,867.8	<0.001	30.1%	36,859.8	<0.001	32.1%	36,745.7	<0.001
Diastolic Blood Pressure	Enterodiol	11.8%	34,721.3	<0.001	12.3%	34,701.3	<0.001	13.3%	34,653.8	0.005
	Enterolactone	11.8%	34,718.8	<0.001	12.3%	34,701.2	<0.001	13.3%	34,655.6	0.007

^1^ *p*-value reflects the level of significance for microbial lignan metabolites, HEI score, and total energy intake as a three-dimensional smooth term for the outcome variable (triglycerides, total cholesterol, low-density lipoprotein (LDL) cholesterol; high-density lipoprotein (HDL) cholesterol; systolic blood pressure; diastolic blood pressure; body fat percentage; glucose; insulin; oral glucose tolerance test (OGTT); hemoglobin A1c (HbA1c)). Akaike information criterion (AIC); percentage of deviance explained (DE) is for the entire model. Body fat percentage was not adjusted for BMI in model 3.

## Data Availability

Data from the National Health and Nutrition Examination Survey are publicly available online at https://www.cdc.gov/nchs/nhanes/index.htm (accessed on 29 June 2022).

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
