# Peer review of "Diet Quality, Microbial Lignan Metabolites, and Cardiometabolic Health among US Adults"

_nutrients, 2023, doi:10.3390/nu15061412_

Round 1

Reviewer 1 Report

In this paper the authors examined the degree to which key microbial metabolites explain the relationship between diet quality and cardiometabolic health using cross-sectional data from the National Health and Nutrition Examination Survey. This is an interesting study and somewhat novel.

My comments are:

Lines 48-60, the paragraph needs to be reduced.

Line 91, “In the present study, we hypothesized that microbial metabolites would partially explain the relationship observed between diet quality and cardiometabolic health.” A mediation analysis needs to be carried out to test this hypothesis.

Line 81, “However, the relationship between overall diet quality, gut microbiota function, and cardiometabolic disorders remains to be defined.” Need to expand this a bit more. Also, referring to ultra-processed foods in the introduction section but also in the Discussion (line 327) confuses the reader as the healthy eating index includes dietary components related to ultra-processed foods but does not follow the NOVA classification system proposed by Monteiro et al. 2016. Maybe replace ultra-processed foods with unhealthy foods.

I would suggest to replace in the title and elsewhere in the manuscript the phrase “metabolites of microbial metabolism” with “intestinal lignan metabolites” as gut microbiota-derived metabolites refer to a large set of metabolites produced i.e. from bacterial metabolism of dietary substrates.  

Did you consider supplement use in your analysis?

 In Table 1, concentrations of enterolactone and enterodiol should be presented before log-transformed. Also, in statistical analysis part you need write that their values were normalized with log-transformation.

How did you deal with missing values of enterolignans and covariates?

Reviewer 2 Report

Overall: This paper examined the association between diet quality and cardiometabolic health with use of GAMs to explore how microbial metabolites impact this relationship. There were significant interactive effects between at least one of the microbial metabolites and diet quality on most of the cardiometabolic markers. This paper was well written. My biggest concern is interpretability of this work and application to clinical practice. Comments related to this and other more minor issues can be found below.

Abstract:

1.      LDL and HDL need to be spelled out. Related to this, the list of cardiometabolic markers on line 31 could be more specific as opposed to more general domains as it is now.

2.      On line 39, maybe after urinary enterolignans, you could put something in ( ) indicating that higher is better. Not all readers will be familiar as to whether higher or lower values of enterolignans are better.

3.      I am not sure what you mean by “strongest support for potential moderating….” on line 39. Do you mean strongest in regards to effect sizes, lowest p-values........etc? Maybe some data could be added to support this statement.

Introduction:

4.      Generally speaking, I think the Introduction was well written and clear. However, I think too much focus was put on aspects of diet not directly related to this work in some sections of the Introduction. For example, there are a few sentences related to plant-based diet and the microbiome. Plant-based diets weren’t studied in this paper. In fact, not all plant-based diets will score well on the HEI because they specifically exclude components of the HEI. The same can be said of the text related to ultra-processed foods. I get it would be very hard to eat a lot of ultra-processed food and score well on the HEI. Having some text (maybe less than what you currently have) related to plant-based and ultra-processed is helpful. However, I wonder if it would help to include more text related to how whole diet approaches have been associated with the gut microbiota function and cardiometabolic health. What has been done with either other indices, Mediterranean diet, or the Dietary Inflammatory Index……etc?  

5.      The last statement of the Introduction is written in a way I would expect to see for mediation. Based on what I understand about GAM, it isn’t a mediation method. In fact, you specifically looked at effect modification. If my thinking is correct, I think you need to change the wording in the last sentence of the Introduction.

Methods:

6.      Did you consider using waist circumference as an endpoint? Waist circumference may be a better indicator of cardiometabolic health than BMI given the focus on central adiposity.

7.      Is there a reference for GAM that you can provide?

8.      Did you take into account the complex survey design with weighting, strata and cluster values for NHANES data?

Results:

9.      On line 204, you indicate statistically significant associations for triglycerides, LDL cholesterol, and HDL, but it isn’t clear what those associations are with. Are they with enterolactone, enterodiol, or both? Readers can figure that out from reading further down, but that specific statement on line 204 could be more specific.

Discussion:

10.  Is “account” the right word on line 322?

11.  To me, it sounds like the last sentence of the first paragraph of the Discussion and the second to last sentence are stating the same thing.

12.  From a research perspective, this study is sound with analytic techniques applied in an innovative way. I understand the topic is complex, but I wonder if the methods are too complex. I am not suggesting running different analyses but is there any way to provide a better synopsis/less complex table for the purpose of the Discussion. Ultimately, this paper may have great research usability, but the clinical usability of this paper is going to be very low. I think researchers are going to have a hard time wrapping their head around the methods and interpreting results. To ask a clinician who does not have as extensive experience in research as researchers do, to interpret and make use of such data will not work. Maybe the point is that there doesn’t necessarily need to be clinical interpretability of this work, but I still think researchers will have a hard time comprehending the interpretation of this work.
